# Synthesis of ultra-high molecular weight homo- and copolymers *via* an ultrasonic emulsion process with a fast rate
Uddhab Kalita [1,2,3], Vianna F. Jafari[1], Muthupandian Ashokkumar [2], Nikhil K. Singha [3] ✉ & Greg G. Qiao [1] ✉

In the forefront of advanced materials, ultra-high molecular weight (UHMW) polymers, renowned for their outstanding mechanical properties, have found extensive applications across various domains. However, their production has encountered a significant challenge: the attainment of UHMW polymers with a low dispersity (Đ). Herein, we introduce the pioneering technique of ultrasound (US) initiated polymerization, which has garnered attention for its capability to successfully polymerize a multitude of monomers. This study showcases the synthesis of UHMW polymers with a comparatively low Đ (≤ 1.1) within a remarkably short duration (~ 15 min) through the amalgamation of emulsion polymerization and high-frequency ultrasound-initiated polymerization. Particularly noteworthy is the successful copolymerization of diverse monomers, surpassing the molecular weight and further narrowing the Đ compared to their respective homopolymers. Notably, this includes monomers like vinyl acetate, traditionally deemed unsuitable for controlled polymerization. The consistent production and uniform dispersion of radicals during ultrasonication have been identified as key factors facilitating the swift fabrication of UHMW polymers with exceptionally low Đ.

Ultra-high molecular weight (UHMW) polymers have found applications in various fields ranging from medical implants, industrial products such as conveyor belts, gears, bearings, etc., in the aerospace industry as parts of engine, wing components and landing gears, in sports equipment to even bulletproof vests[1]. The high molecular weight contributes to the higher strength-to-weight ratio of this kind of polymer. UHMW polymers also show excellent abrasion resistance and better impact energy absorption[2–4]. All these properties make UHMW polymers a sought-after product in the present market. However, attaining a considerably low dispersity (Đ) for UHMW polymers, which facilitates improvement in properties[5,6], has remained a challenge. The reversible deactivation radical polymerization (RDRP) method has been utilized in a few cases under high pressure while using heterogeneous media to achieve low Đ UHMW polymers[7–11]. Recently, enzyme-mediated Fenton-RAFT (reversible addition-fragmentation chain-transfer) polymerization has been employed to synthesize UHMW homo- and co-polymers in aqueous media[12–14]. However, all these methods involve significant use of chemicals and organic solvents, and longer polymerization time to yield UHMW polymers, resulting in more energy consumption during the overall process. Furthermore,

translating these processes into an industrial scale is challenging due to the high cost involved and the maintenance of a very sophisticated environment. We have addressed this challenge by the combination of emulsion polymerization with ultrasonic initiation. As of our current understanding, commercially utilized UHMW polymers are predominantly olefinic. While natural rubber possesses exceptionally high molecular weights, reaching into the millions, it is primarily derived from plants like *Hevea brasiliensis*[15]. The dominance of olefinic UHMW polymers in the market suggests the vast potential for UHMW polymers derived from diverse monomers. However, the limited availability of non-olefinic UHMW polymers has resulted in a scarcity of knowledge regarding their potential applications.

Throughout the past decades, the emulsion polymerization technique has evolved to be one of the most common techniques for the preparation of commercial polymers[16]. There are many advantages of this technique over the other polymerization techniques, *viz.*, use of water in most cases, making it a greener process; easy scale-up, considerably higher molecular weight polymer products, direct use of the emulsion latex without any processing in paints and coatings, and cost-effectivity and so forth. The emulsion polymerization technique widens the scope of monomers that can be

[1]Polymer Science Group, Department of Chemical Engineering, The University of Melbourne, Parkville 3010 Victoria, Australia. [2]School of Chemistry, The University of Melbourne, Parkville 3010 Victoria, Australia. [3]Rubber Technology Centre, Indian Institute of Technology Kharagpur, Kharagpur 721302 WB, India. ✉e-mail: nks@rtc.iitkgp.ac.in; gregghq@unimelb.edu.au

polymerized and offers high conversion rates, resulting in higher product yields, and the products inherently have better processability[17,18]. However, challenges such as careful control of reaction parameters to avoid coagulation exists for this polymerization technique.

Emulsion polymerization using ultrasound (US) has already been carried out by several-research groups[19–23]. The first attempt to use the US to enhance chemical reaction rates was reported back in 1927 by Richards et al.[24], and the use of US in the field of chemical synthesis has since been captivating the scientific community. Acoustic cavitation generated under the effect of ultrasound has the capability to chemically and physically alter reactions. The intense conditions created by acoustic cavitation, where a bubble collapse may yield a localized temperature >5000 °C and pressure of about 2000 atm, may act as an initiator by breaking chemical bonds of molecules generating radicals, and thus enhancing the polymerization rate[23,25]. Polymerization via ultrasound in aqueous media can solve the problem of the involvement of any chemical initiators and organic solvents, as discussed earlier, generating purer polymer products. The disadvantages, though, lie in the case of monomers with very high vapor pressure (such as methyl acrylate, vinyl acetate), in which case the fraction of monomer converted is either very low or no polymerization occurs[26,27]. While most of the previous studies focused on the control of the emulsion particle size and the study of polymerizability of a variety of monomers[28–32], the overall control over the dispersity of the molecular weight for a variety of monomers and their copolymers have not been explored. Bradley et al., in an early work in the year 2002, discuss ultrasonic emulsion polymerization of methyl methacrylate (MMA) and butyl acrylate (BA) in the presence of dodecyltrimethylammonium chloride, a cationic surfactant. They achieved a Đ of about 3 and beyond, and $M_n$ values less than a million Dalton[31]. Though they could not produce a UHMW polymer with low Đ, the study paved the way for using US and emulsion technique in combination as a viable method for polymerizing hydrophobic monomers. Teo et al. compared the use of a non-ionic (Triton-X 100), an anionic (sodium dodecyl sulfate, SDS), and a cationic (didodecyldimethylammonium bromide) surfactant on the stability of the emulsion latexes formed by use of ultrasound-initiated polymerization[33]. It was seen that anionic surfactants provided the best performance when hydrophobic monomer, such as butyl methacrylate (BMA) was being polymerized in emulsion using this process. Synthesis of low dispersity homo- and co- UHMW polymers with high yield within a remarkably short time has also not been achieved to date. Furthermore, majority of the early work on ultrasonic emulsion polymerization had employed a low-frequency US of 20 kHz, where a possibility of degradation of polymer persists due to the high mechanical effects of US at a low frequency[25]. These factors inspired us to exploit high-frequency (HF) US and emulsion polymerization technique utilizing SDS as an anionic surfactant to polymerize hydrophobic monomers and widen the scope further by extending the method to copolymerization of these monomers. It is worth mentioning that radical formation under the effect of US is maximum at ~500 kHz frequency[25], and thus we have utilized the 490 kHz ultrasonic transducer during this study.

The overall process employed here can be termed to be a sustainable or "green" process based on the following points: Firstly, the process utilizes an aqueous-based emulsion system, eliminating the need for organic solvents during polymerization. This not only mitigates concerns about volatile organic compound (VOC) emissions but also streamlines the polymerization process by eliminating the need for solvent removal post-polymerization[16]. Secondly, ultrasound is employed to generate radicals for polymerization instead of chemical initiators like thermal, redox, or photoinitiators. This approach yields purer polymers while reducing the environmental footprint associated with using additional chemicals in the polymerization process.

Herein, we demonstrate the utilization of HF ultrasound-initiated emulsion polymerization of hydrophobic (meth)acrylate monomers to achieve UHMW polymers with a comparatively low dispersity. The polymerization reaction is very fast, and for most of the homo- and co- polymerizations, a full conversion (wt. %) was achieved within 15 minutes. In contrast, conventional polymerization methods take several hours to produce polymers. The homopolymerization of butyl methacrylate (BMA), hexyl acrylate (HA), and methyl methacrylate (MMA) yielded UHMW polymers with a considerably low dispersity (Đ ≤ 1.1). The low Đ of a polymer is usually a characteristic of RDRP processes or anionic/cationic polymerization reactions; however, conventional radical polymerization under the effect of ultrasound combined with the emulsion polymerization technique has been exploited here to achieve the same along with the ultra-high molecular weight of the polymers. Copolymerization of various monomers, including vinyl acetate and superhydrophobic monomers like isobornyl methacrylate (IBMA) and lauryl acrylate (LA) has also been carried out successfully, yielding polymers, in most cases, with higher MW and lower Đ than their respective homopolymers. Dynamic light scattering (DLS) study of the collected latexes for all the polymerization reactions exhibited narrow particle size distribution (PdI) and consistent particle size (Z-avg) for extended periods of time.

## Results and discussion
### Homopolymerization
Homopolymerization of acrylate and methacrylate monomers was carried out in aqueous-based emulsion polymerization using high-frequency ultrasound to produce the radicals necessary for the initiation of polymerization (Fig. 1). Homopolymers of BMA (H1) and HA (H2) were successfully prepared with 100% conversion (wt.%) within just 15 minutes of irradiation time, exhibiting ultra-high molecular weight and comparatively low dispersity (≤1.1) (Table 1; Fig. 2). The controlled and continuous production of hydroxyl radicals under the effect of ultrasonication presumably leads to the production of UHMW polymers in such a short time. It can be implied that the even distribution of hydroxyl radicals throughout the polymerization system generated by means of the collapse of bubbles produced during ultrasonication[25] eventually leads to the dispersity of the polymers being narrow. The US used apparently also controls the growth of the emulsion particles via its inherent mechanical effects when the polymerization is occurring, and thus the Z-avg particle size and polydispersity index (PdI) of the emulsion particle values are very consistent, and no coagulum is observed for the emulsion latexes. It was observed that the emulsion particle size (Z-avg) for these final latexes, after polymerization, is within 50 nm to 70 nm with a very narrow distribution (PdI) (Supplementary Fig. 1, Table 1). The latexes were remarkably stable over a long period, and the DLS study of these latexes after 4 months still exhibited identical Z-avg and PdI values for the emulsion particles (Supplementary Fig. 2). It is worth mentioning that, initially, a dimethylformamide (DMF)-based GPC with RI detector was used to generate GPC trace data for the PBMA before switching to tetrahydrofuran (THF)-based GPC. Both methods showed similar molecular weights and dispersities, but THF's better compatibility with all the monomers led to its continued use.

On the other hand, in the case of MMA monomer, a 100% conversion even beyond 15 minutes of polymerization time (H3) could not be achieved due to its slight solubility in water[34]. MMA thus can be anticipated to have diffused more into the aqueous phase than the other monomers, and it is possible that some of it has been consumed during some side reactions forming water-soluble products[34]. Nevertheless, the MW achieved was relatively high for MMA with a significantly low dispersity within 15 minutes of polymerization. For monomers IBMA and LA, the homopolymers produced (H4 & H5, respectively) also exhibited ultra-high molecular weight; however, the Đ was slightly broader (Supplementary Fig. 3) than those seen for H1, H2, and H3 (Table 1). Both these monomers, IBMA and LA, have bulky pendant groups, and the homopolymerization of these monomers must overcome a certain amount of steric hindrance during the chain growth stage. This very fact possibly leads to the broader dispersity for the polymers of these two monomers having larger pendant groups. Interestingly, Llorente et al. reported that homopolymerization of IBMA, like superhydrophobic monomer in emulsion, is very challenging[35]. Most of the homopolymer latexes of IBMA prepared in their work had a coagulum of more than 15%.

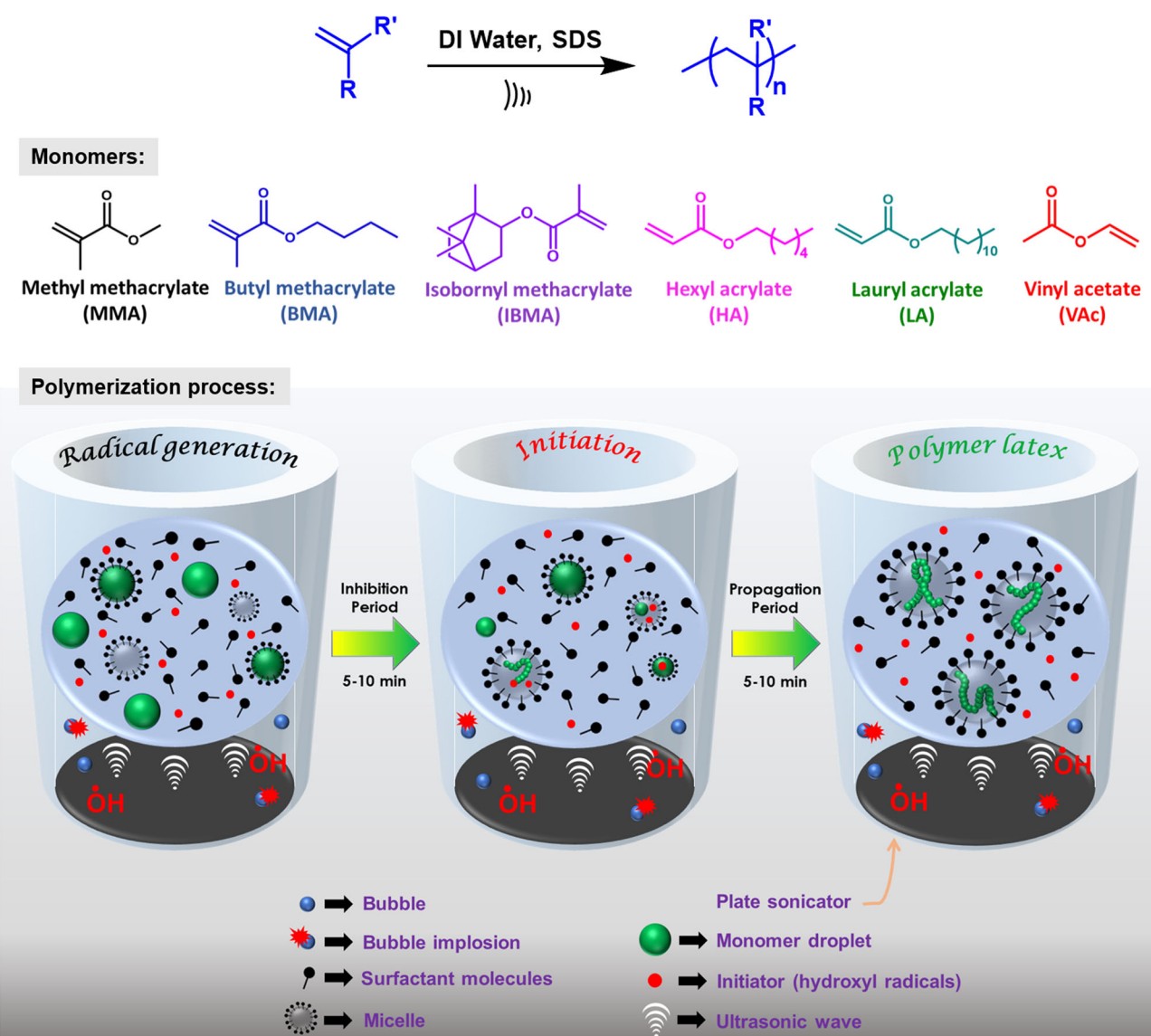

**Fig. 1 | Ultrasound-initiated polymerization of hydrophobic monomers in emulsion.** The structure of the different monomers used, and the schematic representation of the probable polymerization process are shown here.

**Table 1 | GPC, DLS data, and average number of polymer chains per particle ($N_c$) of the (meth)acrylate monomers polymerized using ultrasound-initiated emulsion polymerization**

| Sample ID | Monomer | $M_n$[a] (g/mol) | Đ | Reaction time | Particle size (Z-avg./ d.nm)[b] | PdI[b] (DLS) | Conversion (gravimetric, wt.%) | $N_c$[c] (Chains/particle) |
|---|---|---|---|---|---|---|---|---|
| H1 | BMA | $7.60×10^6$ | 1.01 | 15 min | 54.77 | 0.044 | >99 | 7 |
| H2 | HA | $5.00×10^6$ | 1.04 | 15 min | 67.72 | 0.054 | >99 | 20 |
| H3 | MMA | $3.20×10^6$ | 1.10 | 15 min | 34.04 | 0.347 | 70 | 4 |
| H4 | IBMA | $5.20×10^6$ | 1.37 | 15 min | 51.35 | 0.112 | >99 | 6 |
| H5 | LA | $4.60×10^6$ | 1.43 | 15 min | 70.92 | 0.272 | >99 | 16 |

[a]$M_n$ and Đ were obtained from the Gel Permeation Chromatography instrument. $M_n$ data presented here are polystyrene standard sample equivalents.
[b]Z-avg (d.nm) and PdI were obtained using a Dynamic Light Scattering instrument.
[c]$N_c$ was calculated depending on the weight avg. molecular weight (Mw) of the polymer and the Z-avg size of the latex particle[33,34].

Regardless, a 100% conversion and a UHMW polymer for these monomers were also achieved within 15 minutes of polymerization time. The particle size distribution and size (Z-avg) for the latexes collected for these homopolymers displayed consistency and very stable emulsion over a long period of time (Table 1, Supplementary Fig. 2).

**Copolymerization**

Copolymerization of various monomers was similarly carried out in emulsion using US-initiated emulsion polymerization (Table 2). The same trends of high molecular weight and relatively low dispersity were observed in most of the copolymerization of different monomers. Interestingly, bulky

**Table 2 | GPC and DLS data of the copolymers comprising various monomers polymerized using US-initiated emulsion polymerization**

| Sample ID | Polymer composition[b] | $M_n$[a] (g/mol) | Đ[a] | Particle size (Z-avg./ d.nm) | PdI (DLS) | Conversion (wt.%) |
|---|---|---|---|---|---|---|
| C1 | Poly(MMA-*co*-BMA) | $7.30 \times 10^6$ | 1.001 | 80.64 | 0.060 | 90 |
| C2 | Poly(IBMA-*co*-BMA) | $2.60 \times 10^7$ | 1.04 | 56.31 | 0.064 | >99 |
| C3 | Poly(LA-*co*-BMA) | $1.00 \times 10^7$ | 1.03 | 61.43 | 0.128 | >99 |
| C4 | Poly(VAc$_{20}$-*co*-BMA$_{80}$) | $9.30 \times 10^6$ | 1.003 | 82.26 | 0.163 | 85 |
| C5 | Poly(VAc$_{50}$-*co*-BMA$_{50}$) | $3.90 \times 10^6$ | 1.32 | 72.78 | 0.204 | 65 |
| C6 | Poly(BMA-*co*-IBMA-*co*-LA) | $1.10 \times 10^7$ | 1.11 | 64.73 | 0.182 | >99 |
| C7 | Poly(MMA-*co*-BMA-*co*-IBMA) | $3.60 \times 10^6$ | 1.21 | 59.21 | 0.105 | 85 |

[a]$M_n$ and Đ were obtained from the Gel Permeation Chromatography instrument. $M_n$ data presented here are polystyrene standard sample equivalents.
[b]The numbers 20, 50 & 80 represent the feed ratio (wt. %) of the particular monomer in the polymerization system.

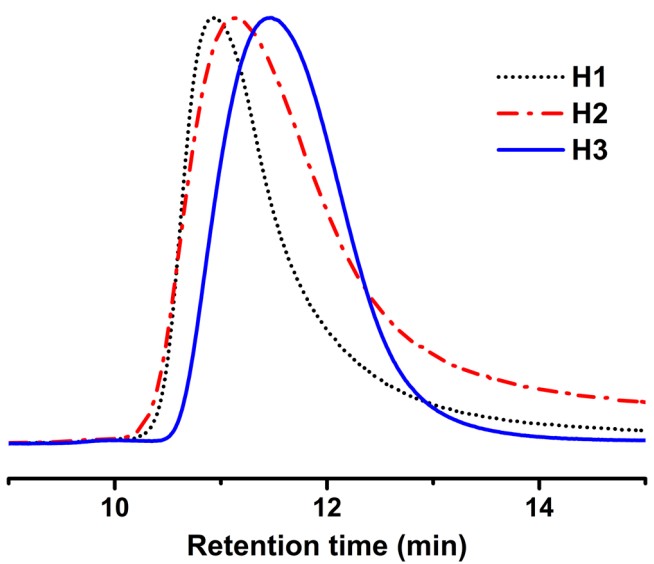

**Fig. 2 | GPC traces of the homopolymers of BMA (H1), HA (H2), and MMA (H3) prepared using ultrasound-initiated emulsion polymerization.** The molecular weight obtained for all these homopolymers is very high ($M_n > 3.1$ million Da) with reasonably low dispersity (Đ ≤ 1.1).

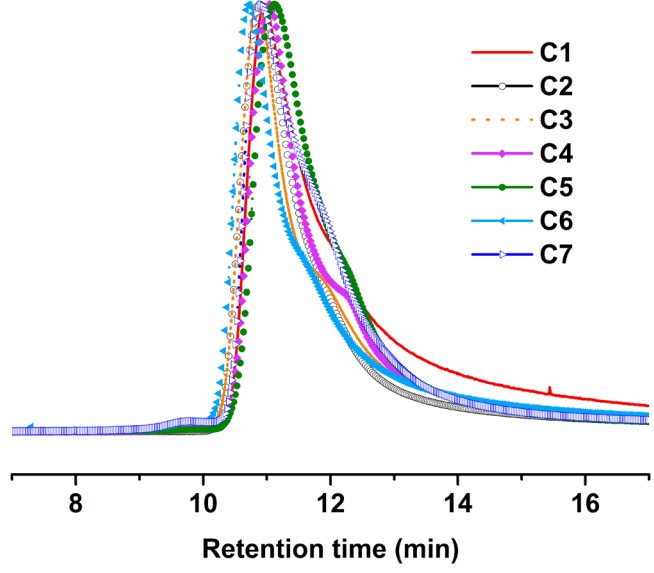

**Fig. 3 | GPC traces of the copolymers of various monomers prepared using ultrasound-initiated emulsion polymerization.** The different copolymers are labelled as follows: C1 = Poly(MMA-*co*-BMA); C2 = Poly(IBMA-*co*-BMA); C3 = Poly(LA-*co*-BMA); C4 = Poly(VAc$_{20}$-*co*-BMA$_{80}$); C5 = Poly(VAc$_{50}$-*co*-BMA$_{50}$); C6 = Poly(BMA-*co*-IBMA-*co*-LA); C7 = Poly(MMA-*co*-BMA-*co*-IBMA). The molecular weight achieved for these copolymers is even greater than their respective homopolymers with a significantly narrower dispersity. The molecular weight of 10-20 million has been breached for these copolymers.

and highly hydrophobic monomers such as IBMA and LA showed a comparatively broader dispersity when homopolymerized using the same method (H4 & H5, Table 1). However, when copolymerized with BMA (50 wt. % of each), the final copolymers exhibited an even higher molecular weight than the homopolymers of the same monomers and a significantly low dispersity. The dispersity for poly(IBMA-*co*-BMA) (C2) and poly(LA-*co*-BMA) (C3) decreased from around 1.40 to 1.04 and 1.03, respectively, as summarized in Table 2 (Fig. 3).

It is known that preparation of very high MW and relatively low dispersity polymer via conventional or controlled radical polymerization involving vinyl acetate (VAc) monomer is very difficult[36]. Thus, VAc has likewise been subjected to polymerization using this method. However, due to the very high vapor pressure as well as its solubility in water, a comparable amount of final polymer could not be obtained. It is worth mentioning that, in previous reports involving polymerization using US, it was established that monomers with high vapor pressure do not allow enough radicals to be generated under the effect of US[26,37]. Essentially, under ultrasonication, the monomers may diffuse into the cavitation bubble/bubble surface. In the case of monomers with higher vapor pressure, they also contribute to higher vapor pressure inside the cavitation bubbles and eventually reduce the intensity of bubble collapse, leading to a lower flux of radicals generated for the initiation of polymerization.

To reduce the effect of vapor pressure of VAc, BMA was also introduced to the polymerization mixture containing VAc in varied ratios of the two monomers. Poly(VAc$_{20}$-*co*-BMA$_{80}$) (VAc:BMA = 20:80 wt.% feed ratio) (C4) polymer preparation was successful (Đ = 1.003); however, a 100% conversion was not achieved. When VAc and BMA were copolymerized at 50:50 wt.% ratio (C5), the MW almost became half of C4, and the dispersity was broader (Đ = 1.32). The successful incorporation of VAc into the copolymer C5 has also been confirmed using $^1$H NMR spectroscopy, as shown in Supplementary Fig. 4 (Supplementary Data 1). Furthermore, copolymerization maintaining VAc:BMA = 80:20 wt. % feed ratio did not again yield considerable amount of polymers for characterization to be carried out, similar to what was observed for homopolymerization of VAc. Moreover, it is known that at very high MWs of vinyl acetate, branching is observed, which is evident from the GPC traces displayed separately for the copolymers in Supplementary Fig. 5(b), exhibiting a slight shoulder for the vinyl acetate copolymers. Copolymerization of three different monomers in a single batch was likewise carried out under the same conditions. The final polymers prepared also displayed very high MWs with a low Đ (C6 & C7; Table 2, Fig. 3).

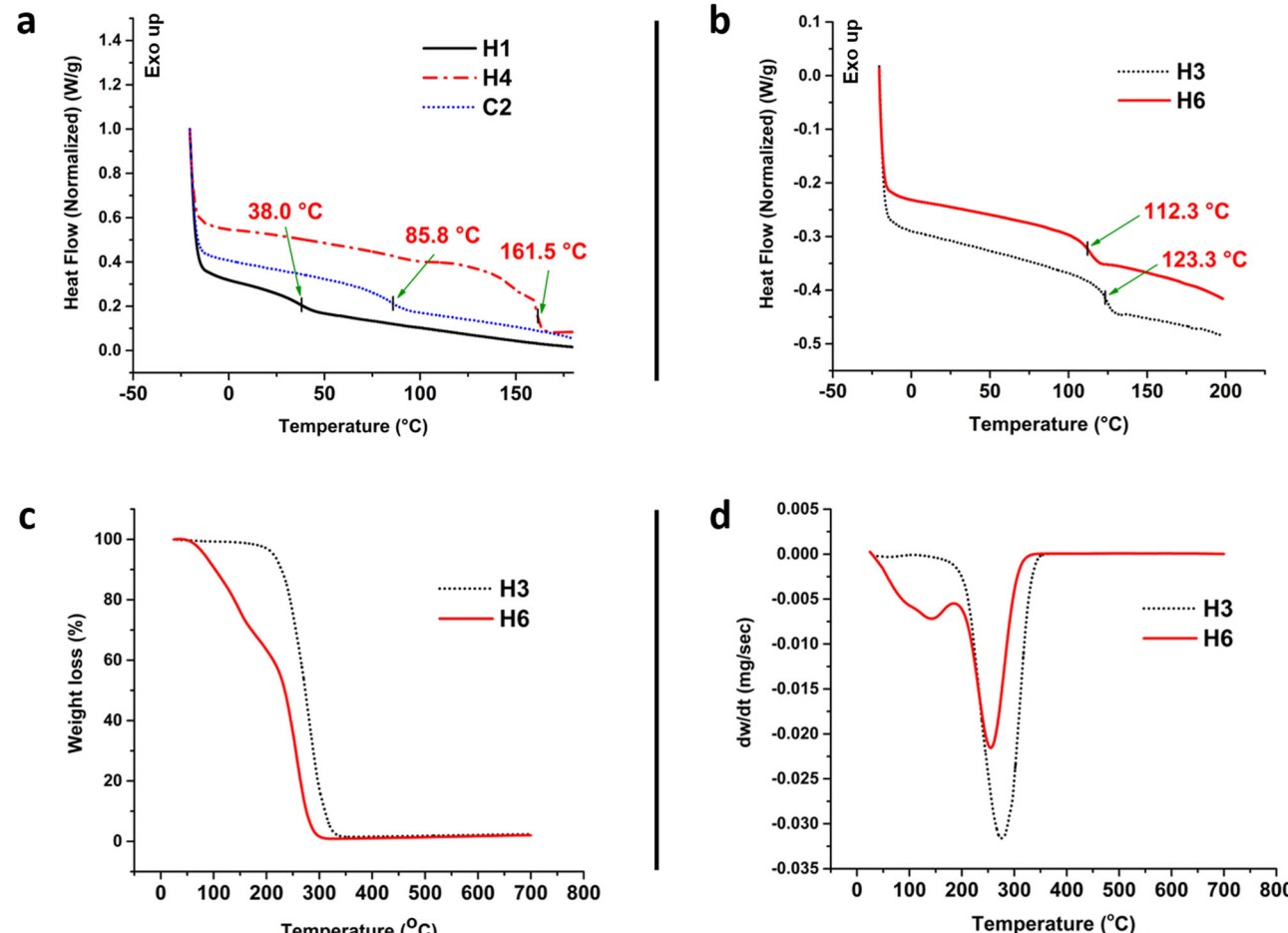

**Fig. 4 | Thermal properties of the prepared polymers *via* ultrasonic emulsion.**
**a** Comparison of DSC traces of PBMA (H1), PIBMA (H4), and poly(IBMA-*co*-BMA) (C2). The single glass transition temperature obtained for the copolymer solidifies that a truly random copolymer has been produced after polymerization rather than forming different blocks or blends of the same monomers. **b** DSC trace comparison of PMMA prepared using US (H3), and PMMA prepared using conventional thermally initiated FRP (H6). The $T_g$ for the UHMW polymer prepared using US is more than that prepared using the conventional FRP. (**c**) TGA and (**d**) DTG thermogram of H3 and H6. The H3 showed better degradation stability than H6 due to the absence of head-to-head linkages.

The latexes collected after each polymerization reaction were characterized using DLS to determine the particle size of the polymers in the emulsion. The size and the polydispersity of particle size for the different copolymer latexes have been summarized in Table 2. Supplementary Fig. 6 displays the particle size distribution of the various copolymers produced. It can be observed that the particle size distribution is very narrow for these copolymer latexes.

### Properties of UHMW polymers and possible mechanism

Differential scanning calorimetry (DSC) analysis was carried out to determine the thermal properties of the homo- and co-polymers. As evident from Fig. 4a, the glass transition temperatures ($T_g$s) for H1 (PBMA) and H4 (PIBMA) homopolymers were 38.0 °C and 161.5 °C, respectively. The $T_g$ for the copolymer of BMA and IBMA (C2), however, displays only a single $T_g$ of 85.8 °C. This indicates that the polymer formed during the polymerization reaction is neither a block copolymer nor a mixture of two homopolymers. Moreover, the $T_g$ value obtained for C2 is in accordance with the calculation of $T_g$ for a copolymer of BMA and IBMA using the Fox equation under these conditions (Supplementary Equation 1). The same phenomenon of having only a single $T_g$ for the copolymer of BMA and MMA (C1) was observed in Supplementary Fig. 7. Whereas the homopolymers H1 and H3 showed $T_g$ values of 38.0 °C and 123.3 °C, respectively, the copolymer C1 displayed a $T_g$ of 64.5 °C. This value of $T_g$ for C1 likewise follows the Fox equation calculation, considering that MMA does not reach a full conversion even during copolymerization. Furthermore, it is observed that there is a marked

difference between the $T_g$ of the homopolymers H1, H3, & H4 (Fig. 4a,b) with the generally reported values for the same homopolymers. The $T_g$ achieved for H1, H3 and H4 increased considerably from 20 °C[38] to 38 °C, 112 °C to 123 °C (Fig. 4b), and 150 °C[39] to about 161 °C, respectively. This phenomenon may be observed due to the ultra-high molecular weight achieved during this work, contributing to the lesser free volume within the polymer and higher chain entanglement of the polymeric chains leading to lower mobility[40].

The hypothesis that under the effect of ultrasonication, the controlled and continuous production of hydroxyl radicals leads to the production of UHMW polymers in a very short period put forward for homopolymers seems to also hold for the copolymers. Moreover, the copolymerization of specific monomers (like IBMA & LA) with a different monomer (BMA) can contribute to the relatively low dispersity of the overall copolymer. This may be because the BMA monomer during copolymerization with IBMA and LA (C2 & C3, respectively) essentially nullifies the steric hindrance faced by the monomers IBMA and LA during their homopolymerization.

Thermogravimetric analysis (TGA) of the homo and copolymers was carried out to analyze the thermal degradation behavior. Figure 4c shows the TGA thermogram of PMMA synthesized using the ultrasonication method (H3) and PMMA synthesized using a conventional free radical polymerization (H6), where a thermal initiator was used to initiate polymerization. The PMMA prepared using conventional FRP (H6) exhibits two-step degradation behavior, whereas H3 only showed a single degradation temperature. The DTG peak (Fig. 4d) at 141 °C appears due to the

scission of head-to-head linkages present in the homopolymer H6, which is absent in the case of H3 homopolymers[41,42]. The ultrasonic initiation leads to the preferential formation of head-to-tail linkages leading to the higher thermal stability of the polymer and, thus, the observation. Interestingly, the single-step degradation temperature observed for H3 at 275 °C is higher than the second degradation temperature for H6 (255 °C) and is probably due to the ultra-high molecular weight of H3.

We hypothesize that ultrasound is able to produce a constant flux of radicals which can initiate polymerization. The polymerization time is generally very fast (~15 min). These two characteristics, i.e., 1) the uniform distribution of the radicals throughout the polymerization system and 2) short reaction time, provide each growing polymer chain within micelles, uniform conditions, producing UHMW polymers with reasonably low dispersity. Rajamma et al.[43]. have quantified radical production (1.2–1.4 μM min$^{-1}$ W$^{-1}$ via Weissler method) at 60 W using 490 kHz frequency, which exhibited a high flux of radicals being generated, along with other researchers demonstrating the successful use of this particular US frequency[26,44]. Moreover, as we maintain a constant ultrasonic power level in the emulsion system throughout all the experiments, we expect that the radical concentration level to remain constant within this short reaction time. To validate this hypothesis, a "semi bio-Fenton" chemistry-based emulsion system was introduced to polymerize BMA. This particular system was chosen because of its supposed ability to supply a constant flux of hydroxyl radicals to the system. While all the ratios of SDS, water, and monomer were kept the same as that used for H1 polymerization, a catalyst system involving ammonium ferrous sulfate (AFS), glucose oxidase (GOx), and D-glucose was introduced for the initiation of polymerization. The final polymer (H7) obtained after purification was subjected to GPC analysis, and the molecular weight achieved was $2.3 \times 10^5$ g/mol with a dispersity of 1.31 (Supplementary Fig. 8). Though the Đ achieved using semi bio-Fenton chemistry was slightly higher compared to the one produced via US and reasonably low compared to the conventional free radical polymerization (FRP) method, the MW attained, however, is not as high as what was achieved using ultrasound-initiated emulsion polymerization. This observation may be due to the instantaneous production of radicals as soon as the AFS is added to the system. The iron (II) from AFS essentially reacts with the hydrogen peroxide produced by the reaction of GOx and D-glucose (Supplementary Fig. 9). In this case, it dissociates $H_2O_2$ to produce hydroxyl radicals necessary for the initiation of polymerization. However, since the amount of oxygen reacting with the GOx/D-glucose system could neither be quantified nor controlled, instantaneous radicals are being generated, or the radicals are not as evenly distributed as in the case of US, eventually reducing the MW of the produced polymer. The other possible reason is that polymerization time is much longer (4 h) under semi bio-Fenton chemistry-based emulsion system and hence difficult to maintain uniform radical flow within the reaction timeframe. Additionally, the identical GOx/D-glucose system was employed for deoxygenation in a reaction system akin to H1 polymerization and underwent initiation using ultrasound for BMA monomer (Supplementary Fig. 10). In this scenario, PBMA (H8) was obtained within an hour, exhibiting a molecular weight comparable to H1 but with a slightly broader dispersity index (Đ) of 1.10. The presence of dissolved gases, especially argon, affects radical flux during ultrasonication, leading to faster polymerization rates compared to enzymatic oxygen removal, resulting in a slower polymerization rate. Nevertheless, with control over the concentration of oxygen in the system or by any other method, which may produce radicals constantly and evenly throughout a polymerization system, UHMW polymers with comparatively low Đ might be fashioned.

## Conclusion

Homopolymers of hydrophobic monomers were successfully prepared using high-frequency ultrasound coupled with an emulsion polymerization technique to produce ultra-high molecular weight polymers with reasonably low dispersity under an argon atmosphere. In most cases, a full conversion (wt. %) was achieved after only 15 minutes of polymerization time. The process does not involve any chain-transfer agent or other reagents needed for controlled polymerization to yield polymers with low Đ (≤1.1). Copolymers of a variety of monomers were also prepared using the same method. It was observed that the copolymerization of two different monomers has a marked effect on the final MW of the copolymer produced. When polymerized, very bulky superhydrophobic monomers or vinyl-based monomers with high vapor pressure, such as vinyl acetate, produced a comparatively broader dispersity polymer or did not yield any polymer. However, copolymerization of this class of monomers with a slightly less hydrophobic (meth) acrylate monomer yielded a higher MW polymer than their respective homopolymers with, again, a reasonably low Đ. It was established that the constant generation and even distribution of hydroxyl radicals under the US is the reason for achieving UHMW polymers with a substantially low Đ in a short time. Furthermore, the possibility of achieving similar MW and Đ using ultrasound-initiation in emulsion without purging with argon or nitrogen has also been successfully explored, utilizing the GOx/D-glucose catalyst system for deoxygenation. This unprecedented finding of ultra-high molecular weight and significantly low dispersity polymers via ultrasound-initiated emulsion polymerization will have far-reaching consequences in polymer science and engineering.

Moreover, there is a lack of commercially available non-olefinic UHMW polymers. Additionally, due to the limited preparation of UHMW polymers from non-olefinic monomers, there is a scarcity of knowledge regarding potential applications for these homo- and copolymers. However, this absence does not negate the potential for UHMW polymers derived from alternative monomers to find commercial utility in the foreseeable future. Instead, the dominance of olefinic UHMW polymers in the market underscores the substantial untapped potential that UHMW homo- and copolymers synthesized from diverse monomers could offer across various applications. Furthermore, while low dispersity UHMW polymers may not suit all applications, they offer advantageous properties, especially for materials requiring high strength and durability. Despite challenges in processing narrowly dispersed high MW polymers, suitable additives or process treatments can facilitate their processing.

## Experimental section

**Materials.** The monomers methyl methacrylate (MMA), butyl methacrylate (BMA), isobornyl methacrylate (IBMA), hexyl acrylate (HA), lauryl acrylate (LA), and vinyl acetate (VAc) were purchased from Sigma Aldrich and were used after removal of the inhibitor if any present by passing through a basic alumina column. Sodium dodecyl sulfate (SDS), Potassium persulfate (KPS), Ammonium ferrous sulfate hexahydrate (AFS), Glucose oxidase (GOx), and D-glucose were also acquired from Sigma Aldrich and were used as received. Tetrahydrofuran (THF), methanol, and deuterated chloroform (CDCl$_3$) were procured from Sigma Aldrich and were used as received. Deionized (DI) water was used throughout the work and was generated by the Milli-Q system.

## Characterization and measurements

**Gel permeation chromatography.** A Shimadzu gel permeation chromatography (GPC) instrument equipped with a refractive index (RI), UV, and a Multi-Angle Light Scattering (MALS) detector was used to measure the molecular weight ($M_n$) and dispersity (Đ) of the prepared samples. The GPC was also furnished with two Agilent PLgel 5 μm MIXED-C columns in series along with DAWN 18-angle MALS detector from Wyatt Technology. THF was employed as the eluent at a 1 mL/min flow rate. The MALS detector of the GPC instrument was calibrated using a polystyrene standard of 30,000 g/mol molecular weight and was eventually used to determine all the GPC data recorded for this work using the ASTRA software (Supplementary Fig. 11 & 12; Supplementary Table 1).

## Dynamic light scattering

Particle size and particle size distribution of the emulsion latexes were determined using a dynamic light scattering (DLS) instrument. A Zetasizer

Nano ZS instrument (Malvern Instruments, UK) equipped with a 4 mW laser at λ = 633 nm operating at a scattering angle of 173° was employed for this study. The latexes were diluted with water before subjecting to DLS measurements. The average diameter, Z-average values (d.nm), and size distribution, i.e., polydispersity index (PdI) values, were recorded using the DTS software.

## Nuclear magnetic resonance

$^1$H nuclear magnetic resonance (NMR) spectra for all the samples were recorded using a Bruker 600 MH$_z$ spectrometer. CDCl$_3$ solvent was used to dissolve the homo- and co-polymers for this study. The generated spectra were analyzed using the TopSpin 3.6.1 software (Bruker).

## Differential scanning calorimetry

Differential scanning calorimetry (DSC) instrument (TA DSC25) was utilized to study the thermal characteristics of the homo- as well as co-polymers. Aluminium hermetic pans were used to mount each sample (5–10 mg), and the polymer-loaded pans were heated at a heating rate of 10 °C/min under the nitrogen atmosphere. For each DSC trace, the $T_g$ was evaluated using the TA instruments' Trios software and was taken at the half height of the transition steps. Origin 2016 was used to plot the DSC traces for representation.

## Thermogravimetric analysis

Thermogravimetric analysis (TGA) for all the prepared samples was carried out using a Shimadzu TGA-50 analyzer at a 20 °C/min heating rate within 30 to 700 °C. Each sample (7–13 mg) was mounted on the designated platform, and Origin 2016 was used to plot the TGA and DTG (Derivative thermogravimetry) data received from the instrument.

## Ultrasonic set-up

During all the ultrasound-initiated emulsion polymerization experiments, an RF generator (AG series amplifier LVG 60-10 manufactured by T&C Power Conversion Inc.) was utilized. The RF generator was operated at an RF applied power of 60 W[43] and was interfaced with an ultrasonic plate transducer (diameter = 5.4 cm) supplied by Honda Electronics Co. Ltd. The transducer, designed to operate at a frequency of 490 kHz, was positioned at the base of a double-walled glass cell ultrasonic vessel. The water bath was fitted with a circulating water-cooling system to maintain precise temperature control and uniformity during the experiments. This system ensured a consistent and stable bath temperature of 25 °C throughout the duration of the polymerization investigations (Supplementary Fig. 13a). For all the experiments, the glass vials with the reaction mixture were immersed in 150 mL of water inside the double-walled glass cell in direct contact with the transducer plate. The temperature of the solution inside and outside the reaction vessel rose from about 28 °C to 37 °C after 15 min of ultrasonication.

## Methods

### Synthesis of homopolymers using US

In a typical polymerization of BMA, 1 wt. % of SDS (0.0447 g) was dissolved in DI water (4.47 g), followed by the addition of the monomer in 5% v/v (0.2 g) with respect to DI water in a glass vial (14 mL) amounting to a total volume of 4.70 mL. The mixture was subjected to 3 pulses of 7 seconds of ultrasonic irradiation of 20 kHz frequency at 30% amplitude to form a stable emulsion. This emulsion mixture was then sealed and purged with Argon for 30 minutes to remove any trace of other dissolved gases, followed by sealing the reaction vial. This vial was then irradiated with ultrasound of 490 kHz frequency at 60 W power for 15 minutes (Supplementary Fig. 13b). After full conversion (wt. %) of the monomer, the irradiation was stopped, and a sample was collected for the DLS study. The emulsion was then disrupted by adding the whole polymerization emulsion mixture to excess methanol. In a few cases, like for monomers such as HA and LA, the methanol used to disrupt the emulsion was slightly acidified. The whole methanol/emulsion mixture was then centrifuged, decanted, and dried in vacuo, and the polymer was then collected for further characterization.

## Synthesis of copolymers using US

The same recipe as that of homopolymers was used to copolymerize various monomers. For a typical copolymerization of BMA and IBMA, 1 wt.% of SDS (0.0439 g) was first dissolved in DI water (4.39 g) in a glass vial. This was followed by the addition of a 0.2 g mixture of BMA and IBMA (5% v/v w.r.t water) in 50:50 wt. ratio. For ter-copolymerization, the different monomers were first weighed out in equal weight ratios, followed by adding the monomer mixture to the SDS/water mixture while maintaining the same v/v of monomer and water. However, during the copolymerization of VAc and BMA, three different wt. ratios of VAc and BMA were employed, with the VAc content being 20, 50 & 80 wt. %. Subsequently, low-frequency US (20 kHz) was used on the final mixture to form a stable emulsion, followed by purging with Argon. The vial containing the emulsion was then subjected to high-frequency US (490 kHz, 60 W) for 15 minutes, and the polymer sample and sample for the DLS study were collected in the same manner utilized for the homopolymers.

## Synthesis of polymers using semi bio-Fenton chemistry and thermal initiation

SDS of 1 wt.% against water was taken in a glass vial containing a magnetic stirrer. Monomer (5% v/v w.r.t. water) was added into the same vial and stirred to form an emulsion. D-glucose (0.1 g) and Glucose oxidase (2 μM) were then added to the mixture. The addition of AFS (1 mmol) marked the initiation of polymerization, and a 100% conversion (wt. %) was achieved within 4 h of polymerization. An emulsion sample after polymerization was collected and diluted immediately for DLS measurements. The emulsion was then disrupted, and the polymer was collected and dried in vacuo, followed by further characterizations.

In the case of conventional free radical polymerization using thermal initiators; SDS, water, and the monomer were retained at the same concentration as in previous experiments. The thermal initiator, KPS, was added in a 1 mol % ratio with respect to the monomer content, followed by purging with Argon. The reaction mixture was then heated at 70 °C under stirring condition. The polymer was collected after full conversion by pouring the whole emulsion mixture in excess methanol and dried in vacuo.

## Synthesis of polymers using catalyst to remove dissolved oxygen followed by ultrasonic initiation

Same amount of SDS, water, and monomer (5% v/v w.r.t. water) were taken in a glass vial as the previous experiments followed by the application of low-frequency US to form stable emulsion. However, instead of argon to remove any other dissolved gases, an enzyme-catalyzed system of D-glucose and GOx was employed into the mixture. A pinch of NaHCO$_3$ was also added to the reaction mixture to maintain the pH of the emulsion. This whole mixture was then stirred and sealed. High-frequency US was then employed on the reaction mixture. After one hour of polymerization, the US irradiation was stopped, and the sample was collected for further characterization as in previous cases.

## Data availability

The article and Supplementary Information contain all the data necessary to support the study's findings and conclusions. $^1$H NMR spectra can be found in Supplementary Data 1. The numerical source data utilized in this report can be found in Supplementary Data 2-4.

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

## Acknowledgements

U.K. is thankful to the Department of Science and Technology (DST), New Delhi, for awarding the DST-INSPIRE fellowship. U.K. and N.K.S. also acknowledge the funding from the Ministry of Human Resource

Development (MHRD), New Delhi, through the Scheme for Promotion of Academic and Research Collaboration (SPARC) program.

## Author contributions

Conceptualization and methodology: U.K., M.A., N.K.S., and G.G.Q.; Formal analysis: U.K; Investigation: U.K, and V.F.J.; Resources: M.A., N.K.S., and G.G.Q.; Writing—original draft: U.K.; Writing—Review & Editing: V.F.J., M.A., N.K.S., and G.G.Q.; Supervision: M.A, N.K.S. and G.G.Q.

## Competing interests

The authors declare no competing interests.
