## [Peer Review File · Communications Chemistry]

This manuscript has been previously reviewed at another Nature Portfolio journal. This document only contains reviewer comments and rebuttal letters for versions considered at Communications Chemistry.

COMMSCHEM-24-0072-T

The authors have responded in part to the comments made by the reviewers and have modified the manuscript. However, some of my original comments still apply and I remain of the opinion that the potential impact is limited and does not warrant publication in a general journal and should be submitted to a more specialist polymer or materials journal. My opinion that "... the results are interesting and may be of interest to the polymer community, they are not particularly surprising in light of literature reports". The work builds on previous literature, including that of the authors. New results are presented in that high molecular weight polymers are produced from a range of monomers but this lack the novelty to have high impact.

For other comments:

1. The methodology is generally sound and adequately described – with one or two notable omissions. For example, the reaction system (Fig S8) is contained in a vial submerged in water held within the reactor. We are told that circulating cooling water kept the temperature constant. However, we are not told what this temperature is or whether it was actually measured inside the reaction or simply in the circulating bath. The value is needed to rule out the (unlikely) possibility of thermal initiation. Was a blank experiment performed to rule out the (also unlikely) possibility of photochemical reaction? Likewise, the applied power was 60W from the transducer. However, this tells the reader nothing about the ultrasound intensity inside the reaction vessel.

Response: The temperature of the circulating water bath was maintained at 25 °C which has now also been incorporated into the Revised manuscript (**Page No: 15; Line No: 441**). However, the temperature inside the reaction vessel was not measured during the reaction. With the current setup, we could not measure the temperature inside the reaction vessel at a given time under the same conditions as that used for the polymerization reaction.

The frequency inside the reaction vessel will not change though there might be some difference between the applied intensity and the intensity experienced inside the reaction vessel. However, it is not possible with the current methods to record the intensity inside the reaction vessel, which even if it decreases will be negligible, as the distance between the sonicator plate and the reaction mixture is within 5-6 cm only. Moreover, our previous works have already well-established the use of this particular high frequency ultrasound²⁻⁴ for various polymerization reactions under varied conditions with success while maintaining about same temperature of the bath. Hence, the same conditions were utilized during this work without further alteration.

Furthermore, we respectfully note that we did not fully comprehend the term "blank experiment" as used by the reviewer. However, we diligently allowed the reaction mixtures to stir for several hours at room temperature and under exposure to natural light, without employing ultrasound. Unfortunately, no polymers were obtained.

The lack of knowledge of the temperature (and less importantly the intensity) inside the reaction vessel is a major weakness in the description and may limit reproducibility by other scientists. The distance from the transducer/emitter may be short but is the reaction vessel acoustically transparent? Would the vessel reflect a proportion of the ultrasound? A blank/control experiment

conducted by sonicating an amount of water for the appropriate time under the same experimental conditions and measuring the temperature immediately after turning off the power would give some indication as to whether the temperature changes.

2. The Abstract (line 31) suggests that ultrasound initiated polymerization represents a “breakthrough” technique. The method was suggested in the 1930’s and studied in detail in the 1950’s and again later with newer characterization techniques being applied. “Breakthrough” is not really justified. Some of the language used is a little unscientific and I do not think this helps ‘sell’ the results.

Response: Our initial use of term “breakthrough” was referenced to the ultra-high MW formation in rapid speed for a variety of monomers using the ultrasound. Nevertheless, the term “breakthrough” has now been removed from the abstract. The abstract has also been rephrased to remove any ambiguity.

Acknowledged

3. The introduction suggests several uses for UHMW polymers. However, in the main (not exclusively) those listed refer to polyolefins – which are not studied in this work. Can the requirement of UHMW versions of these (co)polymers be justified? Likewise, low polydispersity is sometimes, but not always, advantageous in polymer applications.

Response: The Reviewer rightly pointed out that only olefinic UHMW polymers are mentioned in the introduction segment of the manuscript. This is because, to our knowledge, only olefinic UHMW polymers are currently commercially used. Natural rubber does possess very high molecular weights, a few millions, but it is naturally produced by plants such as *Hevea brasiliensis*. However, we have not found any commercially available non-olefinic UHMW polymers. Nonetheless, this absence does not discount the potential for UHMW polymers derived from alternative monomers to find commercial utility in the future. Additionally, the fact that only olefinic UHMW polymers are currently commercially available underscores the vast potential that UHMW polymers synthesized from other monomers may have in numerous applications. Furthermore, as UHMW polymers from other monomers have not been prepared with such ease to date, there is a dearth of knowledge about the potential applications for these homo- and copolymers. Moreover, while not explored in this manuscript, the polymerization of monomers similar to acrylonitrile using this method might open new avenues in the field of polymerization.

Additionally, UHMW polymers with low dispersity might not be a choice for some applications, as the Reviewer pointed out, but it is an advantageous property to have, especially when a material with high strength and better durability is being sought. It is known from the literature that polymers with high MW have excellent mechanical properties and are sometimes difficult to process, especially when they are narrowly disperse. Nevertheless, with suitable additives or process treatment, they can be processed without much difficulty. However, after considering the comments received from the Reviewers, we are not claiming the polymers to be very narrowly disperse, but rather, comparatively narrow disperse with respect to the conventionally prepared polymers.

We have now added a few sentences regarding the same in the **Introduction (Page No: 2; Line No: 61-67)** as well as **Conclusion section (Page No: 14; Line No: 377-387)** of the Revised Manuscript.

It is worth mentioning that, the homopolymers were synthesized to exhibit the suitability of this process for a variety of monomers. The (co)polymers have been synthesized to show that the process can be tinkered with and that the right combination of monomers can give rise to comparatively low dispersity UHMW polymers even though their homopolymers probably had a significantly broader dispersity.

Acknowledged. I agree with the final paragraph. However, showing that UHMW polymers can be made in case some applications can be found for them limits the impact.

4. One of the suggested justifications (lines 65, 66) for the current work is the cost and difficulty in applying synthesis methods for UHMW polymers on an industrial scale. However, there is no consideration of the potential costs or usefulness of using the ultrasound method on a large scale. On a related point, the title of the Manuscript includes the phrase “ A Sustainable Synthetic Approach”. However, there is no discussion of energy consumption, carbon costs or material considerations etc. to justify the sustainability argument.

Response: As rightly pointed out by the Reviewer here, an extensive cost-effectivity evaluation of the proposed method has not been carried out. Furthermore, scaling up this process can also be challenging regarding the commercial viability of the products. But, in this work, our focus is primarily on highlighting and proposing a new method to produce UHMW polymers with comparatively low dispersity, which was not previously achieved with such ease. Notably, the time required for all the polymerization reactions to reach a complete conversion is a meager 15 minutes, which essentially cuts down the cost involved by several folds as the established methods require a very long time to synthesize UHMW polymers.

The term “A Sustainable Synthetic Approach,” as quoted in the title of the Manuscript, was utilized because of the “green” nature of the overall process proposed here. We are highlighting the process as a “green” process depending on the following points:

- i) The process described does not require organic solvents during polymerization and has employed an aqueous-based emulsion system. The use of aqueous emulsion reduces the concern of VOC being generated,⁵ as well as removes the tedious process of removal of organic solvents after polymerization.
- ii) In this work, ultrasound was used to generate the radicals necessary for polymerization instead of chemical initiators (thermal, redox, photo, etc.), eventually leading to purer polymers and reducing the environmental impact of using more chemicals during the polymerization process.

Nevertheless, as per the suggestion of the Reviewer, we have decided to change the title of the Manuscript to omit the term “A Sustainable Synthetic Approach.”

Furthermore, a new paragraph related to the sustainability of the process has also been added in the revised manuscript in the **Introduction** section (**Page No: 4; Line No: 114-122**).

Just because it is rapid does not mean that it is cheap. Running a sonicator for 15 min may use more energy than a 60 °C oven for an hour – this needs to be considered and discussed before any conclusion can be drawn as to potential economics. Can the process be scaled up sufficiently to be interesting to industry? These monomers can break down to other products even in 15 min so the claim of “purer” polymers needs to be substantiated.

5. The background acknowledges some of the previous work in ultrasonic emulsion polymerization but references only one review and two other papers. There is a large body of work – admittedly over a range of different frequencies and powers – that should be acknowledged.

Response: We thank the Reviewer for this suggestion. A few more relevant references have been included in the revised Manuscript for the text regarding ultrasonic polymerization, which was already a part of the introduction section.

Acknowledged.

6. A number of papers have previously referred to narrow polydispersity in particle size. It is also well known from the polymer literature that emulsion polymerization produces high (often much higher) molecular weights than bulk or solution reactions so that the comparison here is not valid. The values in Table 1 of M_n should be acknowledged as ‘polystyrene equivalents’ or details of the single standard calibration included.

Response: As rightly mentioned by the Reviewer, there have been reports where low polydispersity for particle size has been prepared using various methods. However, our intention in showcasing the narrow polydispersity in polymer particle size after using ultrasound was to show that the final latex particles were of even size, which may be correlated to their molecular weight being similar, thus narrowly disperse.

Emulsion polymerization does produce higher molecular weight polymers as compared to bulk or solution polymerization techniques. However, as per our knowledge, ultra-high molecular weights achieved using free radical emulsion polymerization in this work have not been achieved using other free radical emulsion polymerization methods where redox or thermal initiation was employed.

Furthermore, as per the suggestion of the Reviewer, ‘polystyrene equivalents’ term has now been added to Table 1 (**Page No: 6; Line No: 172-173**) and Table 2 (**Page No: 8; Line No: 217-218**) as **footnotes for the M_n .**

Acknowledged. The comments of another reviewer are also relevant to this point.

7. The suggested mechanism of the process seems convincing and is consistent with previous literature. Additional weight could be added if the authors had spectroscopically measured the rate of radical production in the reaction vessel and demonstrated that it was high and constant.

Response: Rajamma *et al.*¹ have already evaluated the amount of radicals generated by ultrasound at a frequency of 490 kHz at 60 W, and we have cited it in the Materials section. Several researchers have well documented the use of this high-frequency ultrasound of 490 kHz, e.g., Collins *et al.*²; Piogé *et al.*³ As we maintained a constant ultrasonic power level in the

emulsion system throughout all the experiments, we expect the radical level also to be constant within the short reaction time. We have also added a description about the radical concentration citing these references in the revised manuscript (**Page No: 12; Line No: 319-324**).

Acknowledged. However, the text merely says that it has been measured and does not give quantitative data.

Reviewer #2 (Remarks to the Author):

The authors have made sufficient revisions and satisfied responses to reviewers including Reviewer 3 upon the last version to Nat. Commun. It can now be publishable in Communications Chemistry.

Reviewer's Comments and their Response

We thank the Editor and the Reviewer for their valuable comments and for the opportunity to revise the Manuscript addressing these comments. Below, we have included the new comments from the Reviewer to be addressed along with the rebuttal for the same for your kind consideration.

Reviewer's Comment: The lack of knowledge of the temperature (and less importantly the intensity) inside the reaction vessel is a major weakness in the description and may limit reproducibility by other scientists. The distance from the transducer/emitter may be short but is the reaction vessel acoustically transparent? Would the vessel reflect a proportion of the ultrasound? A blank/control experiment conducted by sonicating an amount of water for the appropriate time under the same experimental conditions and measuring the temperature immediately after turning off the power would give some indication as to whether the temperature changes.

Response: We thank the Reviewer for this suggestion. We have now measured the temperature of the water/SDS solution after ultrasonicated for 15 minutes under Argon and air atmosphere. The temperature of the solution inside as well as outside the reaction vessel rose from 28 °C to about 37 °C in both cases. We have also included a sentence in the Experimental section of the revised manuscript to disseminate the same information to the reader (**Page No: 12; Line No: 398-400**).

Reviewer's Comment: Just because it is rapid does not mean that it is cheap. Running a sonicator for 15 min may use more energy than a 60 °C oven for an hour – this needs to be considered and discussed before any conclusion can be drawn as to potential economics. Can the process be scaled up sufficiently to be interesting to industry? These monomers can break down to other products even in 15 min so the claim of “purer” polymers needs to be substantiated.

Response: Our claim about the polymers being “purer” referred to the fact that we have not used any other chemical initiator during the process. The chemical initiators are known for their potential safety hazards which in this work has been replaced with water, the most sustainable chemical ever. Moreover, if chemical initiators do not get fully consumed during a polymerization reaction (residual initiator), they may remain trapped in the polymer, affecting the stability, toxicity, and other properties of the products. For scaling up issue, it is not the concern of this manuscript as it is trying to describe a chemistry observation. However, there are plenty of article in the literature about the scaling up issue of ultrasonic technology.

Reviewer's Comment: Acknowledged. However, the text merely says that it has been measured and does not give quantitative data

Response: We have now included the hydroxyl radical generation data reported by Rajamma *et al.*¹ in the revised manuscript. The value for OH• yield measured by the Weissler method was 1.2–1.4 $\mu\text{M min}^{-1} \text{W}^{-1}$.¹ (**Page No: 8; Line No: 273**).

References:

1. Rajamma, D. B., Anandan, S., Yusof, N. S. M., Pollet, B. G. & Ashokkumar, M. Sonochemical dosimetry: A comparative study of Weissler, Fricke and terephthalic acid methods. *Ultrason. Sonochem.* **72**, 105413 (2021).